# Value Creation and Capture with Big Data in Smart Phones Companies

**Maniyassouwe Amana** *, **Pingfeng Liu and Mona Alariqi** *

School of Economics, Wuhan University of Technology, Wuhan 430070, China
* Correspondence: amanamaniyassouwe@gmail.com (M.A.); ariqimona@gmail.com (M.A.)

**Abstract:** With the advent of social media, the volume of data generated on the internet has exploded due to the growing number of social network users and their interactions on the internet. Given that, in the age of the digital economy, data has become raw material in terms of decision making, it is important and urgent to conduct a study to understand the effect of big data on value creation and value capture. The goal of the current research is to study the share of big data in value creation and capture in the companies Apple and Samsung. The main question addressed by this article is whether the increasing volumes of data in the digital age can improve the creation and capture of value. To achieve this goal, we considered active users of the three main social networks that Samsung and Apple companies use for their advertising to describe "big data". We measure the "value creation" through the hardware component of the smartphone, such as the battery, camera, CPU speed, RAM (Random-Access Memory), screen size, and storage. Profit, satisfaction, and unit of phone sold are the three manifest variables considered to measure "Value capture". As a methodology, we used partial least squares and structural equation modeling to obtain the results. The pattern enables us to measure the effect of big data on value creation and value capture. The results indicate that CPU speed, RAMs, and battery capacity are the principal variables that impact the "value creation" in terms of customer need and satisfaction.

**Keywords:** smartphones; big data; value creation; value capture; structural model





## 1. Introduction

Big data is an expression that describes large volumes of data [1–3]. These data are both structured and unstructured. In other terms, it is a collection of data that is large in volume, growing exponentially with time, and complex to analyze with traditional techniques [4]. Data analytics is important in terms of value creation and capture because data help companies with good decision making [4]. Thus, many phone firms are turning to big data that is generated through the internet, in the age of social media, imposed by the new trend of the forth industrial revolution, in order to profit from it [5], because big data makes it possible to improve new product development by taking into account the suggestions of consumers. The suggestions made by customers are contained in the data. Big data assist firms in decision-making and improving customer's product by reducing the cost [6]. With the role and importance of big data, many companies are increasingly interested in the adoption of big data in order to profit from its advantages in terms of value creation. Thus, Big data is increasingly becoming a reference tool for businesses to innovate their business model [7]. This innovation emanating from big data makes it possible to create more economic value [8]. The author has addressed the question concerning "creating and capturing value from big data". His study focuses on creating value by taking advantage of big data at the age of new technologies, in which data become a raw material. Indeed, many companies indirectly collect large amounts of data, regarding comments and suggestions on their advertising platform, with the advent of new information and communication technologies (ICT) and social media. Thus, each company

suddenly becomes a source of massive data [9]. Big data technologies make it possible to analyze unstructured and complex data. Analyzing the comments on their social media platform helps to make better decisions regarding the type of innovation to make [3]. The creation of value makes it possible to establish a commercial system which makes it possible to transform capital or raw material into a finished product [10]. Value creation strategies take into account the product's value, as well as the value of input from shareholders who want their stake increased in value. Big data brings a panoply of opportunity in term of value creation [5].

The research conducted thus far has focused on the framework and theory of big data, its applications, and its dimensions. The studies carried out highlight the importance of big data for companies, and the role of big data in the creation of value. Other study results claim that in order to create value, companies must put in place technologies to exploit large volumes of data. Studies have been conducted to understand how big data can create value. However, no study has been conducted to measure the relationships that exist between big data, value creation and value capture within phone firms. It is urgent and important to do this study in order to understand the effect that big data has on the creation of value through good decision-making.

In this research, we study the impact of big data on value creation and capture and the effect of value creation on value capture. To achieve this goal, data from social media regarding Samsung and Apple companies will be used. The data from the top three social media used by Apple and Samsung for advertisement are considered as big data [11,12]. Indeed, Samsung has more than 330 social media profiles, including 144 Facebook profiles, 98 Twitter profiles, and 64 YouTube profiles. Apple has at least 40 social media profiles, including 10 Facebook profiles, 15 YouTube profiles, and 9 twitter profiles. Social networks are mainly used by smartphone manufacturing companies to reach their target population [13]. Thus, the purpose of this research is to examine how social network users help Apple and Samsung to create and capture value at the age of new technologies, in which people increasingly use social networks to interact. The goal is to measure the effect of the number of subscribers on the Facebook, YouTube and Twitter pages of those companies on the creation of value and the value captured. Value creation, in this study, refers to the hardware components of smartphones, such as the size of the screen, the speed of the processor, the RAM, camera, battery capacity, and storage. Value capture includes customer satisfaction, the phone firms' profit, and the unit of phones sold. In summary, the main question of this research is to study the effect of the social media subscriber's numbers on the value creation and capture of the phone companies Samsung and Apple in the age of social media. In other words, we pose the question: does having a large volume of data in the digital age allow companies to create and capture value?

The objective of this study is to use partial least square structural equation modeling (PLS-SEM) to find the link and correlation that exist between the different latent variables, namely, big data, value creation, and value capture. PLS-SEM is the statistical tool used to analyze latent variables. PLS-SEM assesses the relationships (i.e., their strengths) between latent variables and determines how well the model explains the target constructs of interest. The main reasons for, and advantages of, using PLS-SEM are its ability to estimate very complex models and its low data requirement.

The main contribution of the research is to measure the impact of data from social media user's interactions on the value creation and value capture of Apple and Samsung companies. This research has industrial value. Indeed, the telephone industries, notably Apple and Samsung, seek to determine the factors contributing to their economic growth. Among these factors, the characteristics of smartphones remain decisive [14]. This study enable the determination of which material characteristics of the smartphone induce more value creation among customers. Value capture is conducted through the number of smartphones sold and customer satisfaction [15].

This study also enables the measurement of the manifest variables that compose "Big data", "Value creation", and "Value capture". As sustainable development is the pillar of

new strategies and competitive companies, this study will look into this aspect. Smartphone companies are in competition when it comes to selling their product. To stay the course, they improve the characteristics of their product from the consumer's point of view. This allows for a larger target population [16].

The rest of the work is organized as followed. The description of data collection and the methodology of our study will be given, respectively, in Sections 3 and 4. Section 5 presents the results of the research. However, before that, we conduct a literary review of the work already carried out by other scholars in this direction in Section 2. The conclusion will give the epilogue of this research in Section 6.

## 2. Literature Review

This section presents the existing work of previous researchers in order to review the work that has already been conducted in this field. This literary review will focus on the effect of big data on value creation and value capture. The literary review will also discuss terms such as value creation and value capture.

### 2.1. Big Data Effect on Value Creation

In recent years, the exploitation of big data is recognized as an important lever for boosting businesses as data from big data makes it possible to make a forecast in the supply chain. Wafula et al. [1] studied the supply chain system of Kenyan companies with regard to decision-making. Design Science research is the methodology used to obtain the results of his research. The results indicate that big data makes it possible to carry out a predictive analysis which will therefore have a positive impact on decision-making. In the same logic of thought, Faroukhi et al. [17] have conducted a study on the big data value chain. They based this study on big data value chain models in order to extract the information hidden in these large volumes of data. His research explores the big data-driven value chain framework. The framework makes it possible to monetize big data and thus create value. In his research using a smart big data framework for knowledge discovery, Siham Yousf [18] studied a framework that takes into account data from several sources. Considering data from multiple sources improves the quality of data analyses. This is possible because the framework takes into account the veracity and variety of data. In his research on the impact of big data on the new business model, Liang [19] targeted 318 Chinese companies as a sample to study the impact of big data. Technical talent linked to big data has a positive effect on business model innovation. Ricardo Chalmeta [20] focuses his research on using big data to promote sustainability in supply chain management. The paper proposes a framework for sustainable supply chain management that consists of six dimensions: methodology, organization, stakeholders, maturity model, human resources and technology. The framework combines quantitative and qualitative assessment methods for sustainable development. The framework describes the techniques and technology used in each task of the method. Hassan Keshavarz [21] has carried out research regarding big data analytic value in telecommunication industries. In his research, he finds that big analytical data is beneficial for some companies that understand the challenge linked to this technology. He notes that only a few companies make use of technologies that refer to big data analytics. In his research on the relationship between big analytical data and business performance, Parisa Maroufkhani [2] focused his research on a systematic review. He reviewed articles in the Web of Science catalog. The study identifies the factors that can influence the adoption of big data analytics in different parts of an organization and classifies the different types of activity that big data analytics can address. In view of the literature review on the work carried out by other researchers in the field of big data and value creation, we find that none of the existing research has studied the relationship between big data and value creation and capture in smartphone companies such as Samsung and Apple. The main goal of this research is to fill this gap. In other words, this article's goal is to study the relationship between big data, value creation, and value capture regarding smartphones made by Apple and Samsung.

## 2.2. Value Creation

What are the elements that are involved in creating value? This question is important because businesses should lay more emphasis on the factors that lead to longer-term value creation [10]. However, before that, the definition of certain terms from the point of view of previous research remains important. There is no universal definition of value creation, but we can refer to eminent researchers regarding the creation of value. As part of his study of value creation and capture, Lepak et al. [22] asserts that value creation is the process that allows a company to create value. In their study on the comparison between value creation and value capture, Kraaijenbrink et al. [23] claim that value creation is a term that refers to how to create value. It can be created by companies' members. The members of a company are important drivers in the creation of value. In his research on the perception of customers with regard to value creation, Priem [24] provides a definition of the latter. He asserts that value creation is an innovation that brings satisfaction and increases consumer evaluation.

Dyer et al. (2018) [25] conducted a study on the factors that promote value creation and value capture. From this study, they arrive at a definition of value creation. They claim that the creation of value is the value created in an alliance that is superior to the commercial relationship.

In classical theory, value is created if the goods and services price is greater than the cost of the things used to produce them. The cost of things used refers to the price of resources [26]. In the classical model, the value generated is the amount of surplus that comes from the buyer and producer. Customer surplus is the gap between the best price buyers would be willing to pay for a service and the price they actually pay, while producer surplus is the gap between the value at which sellers are selling and the cost of the resource, they used to produce those goods or services. Therefore, this question of surplus boils down to finding the optimal point that suits the seller and the buyer. Kraaijenbrink et al., and Porter et al. [23,27] conducted a study on value-driven vision of strategic work. They find that, in general, companies do not only compete with similar companies in the sector, but they also compete with their customers. In order to retain customers, based on the definition of the competition, they bring out the different value-oriented notion of competition. It emerges that the fundamental element in terms of economic value creation remains the implementation of the strategy. The strategy involves several factors, such as social mechanisms, which include the policies of governors and qualified human resources.

Wikström [8] claims that shoppers determine prices by setting differentiated gradable objectives that embody the identification of resources and the capability of shoppers before selecting products and victimizing them [28]. The aim of every organization is to build value or add value. Therefore, when an action is taken to create gains that outweigh the costs, or if we avoid taking an action when the costs exceed the actual benefits, value is generated for the company.

In their research on new economy competition, Nanda et al. [29] assert that new value is generated when a company develops or creates new ways of doing things, using new strategy and new technologies. New value is only possible through innovation, improving upon something that has already been conducted, or changing the way an existing technology is used in order to achieve a given purpose. Urbinati et al. [8] conducted a study on big data to understand how companies can benefit from it in terms of value creation. Their results, based on a theoretical framework of value creation and value capture, indicate that big data offers two main strategies for innovating services. These are the case-oriented strategy and the process-oriented strategy. Kullak et al. [30] studied business models based primarily on social networks. They understand that companies with little means can combine their strength in terms of the integration of resources and the creation of shared value for the benefit of all. Müller et al. [31] studied how big data can help create value in Danish industries. They argued that to create value through big data, companies must put in place the technology to exploit large volumes of data and thus

benefit from it. Takahashi et al. [32] conducted a study on the value creation model using case-based decision theory. The results indicate that three elements determine the value of a service: the service, providers, and the receivers. In view of this literary review on the creation of value with big data, we realize that there is not yet a study that measures the effect of big data on the creation of value in smartphone companies. One of the objectives of this research is to fill this gap. This study will measure the impact of big data in the creation of value in smartphone companies.

*2.3. Value Capture*

In their comparative study between value creation and value capture, Bowman [33] asserts that value capture is determined by seller and buyer perception. In their research on the factors that promote value capture, Dyer et al. [25] indicate that value capture is the percentage of value that is affected by each of the partner. Value is divided into three levels: the individual level, societal level, and organizational level [22].

Olszak et al. [34] studied the creation of value based on big data in organizations. They claim that one can capture value through data analysis. We can capture the business valued by big data in a company if the latter implements this technology.

From the points developed, we see that it is essentially the definition of the terms value creation and value capture. The papers we have explored bear on the elements that contribute to the value creation and value capture. How do companies manage to create the value of their products through big data from their advertisement social media? Some scholars underline the creation of value as being induced through the transition from business model to contract, which brings together several actors.

However, it remains to explore how the introduction of a process oriented strategy can boost their service. This strategy consists of involving customers at all levels of the project. This avoids the situation of asymmetries. In future work, we have to deepen our research to understand the dynamic and creative processes that forge the interactions of supply-demand that occur in the process that lead to creating and capturing value through big data generated by internet users.

By following the evolution of big data over a few decades, and underlining the importance of data analytic in terms of structured and unstructured data processing, Lee et al. [35] used merchants' review data to demonstrate the application and the role of data analytics. Data processing enables us to measure the impact of big data on business achievement. They invite researchers to focus more on practical research. Only practice will enable us to meet the challenges related to big data within companies, as well as in industries. All this goes through the establishment of solid infrastructures and the human resource qualified in these types of technologies. In view of these gaps observed in the work of other authors, we propose to fill this gap. The goal in this research is to examine the value creation and value capture with massive data in Samsung and Apple companies and to focus on value creation and capture from smartphones.

## 3. Hypotheses of the Research

As the goal of this research is to establish the relation between big data, value creation, and value capture, in this section, we are going to assume some hypotheses between those different variables. These assumptions are based on the statement of other researchers.

**Hypothesis 1:** *There is a positive correlation between big data and value creation. The larger the data, the more value is created.*

Indeed, data becomes raw material in the production of value. Big data is now an important resource in terms of value creation because skills related to big data can promote the creation of value in the era of the digital economy [19]. This hypothesis is justified because there other scientists have already made this claim. Indeed, Dekimpe et al. [36] claim that big data offer opportunities to create value. The digital economy report examines

the role of digital economy for developing countries in term of value creation. One of the two main drivers of value creation is digital data [37]. Big data can be used to change the product [8]. Indeed, we can analyze customers' data to find out their level of satisfaction and then add value to meet their expectations. Collecting more data from customers enables the creation of more value [38]. The expansion of the digital economy creates value through digital data. This is possible because of the data generated. Through data, we can discover which component of smartphones interest more customers. Big data is important in the process of value creation [39] because we can analyze data for good decision making. This enables the creation of new products or the improvement of existing products. Big data often contribute to value creation in companies [40]. Big data makes it possible to improve the efficiency of a product and save a lot of energy [41] because big data has great impact on business achievement [35].

**Hypothesis 2:** *There is a strong relationship between big data and value capture.*

The more data available, the more value is captured. Indeed, value capture is associated with data capture [42]. Three manifest variables are considered to describe value capture. These include customer satisfaction, company profit from smartphones sold, and the unit of phones sold. Companies sometimes ask their customers to offer value creation by stating the features they require in a phone. The proposals collected from customers, in the form of digital data, make it possible to satisfy the taste of customers in terms of manufacturing smartphones [8]. Comments in the form of photos, text and audio from internet users on social networks allow smartphone companies to improve smartphones by taking into account customer suggestions. Big data allow companies to maximize profit [38].

**Hypothesis 3:** *Value creation has a positive impact on value capture.*

The more value we create, the more we capture. Empirically, we know that if we want to capture more value from a product, we need to take good care of the product when creating it. The value captured is determined by customer's perceptions of a given product [33]. The perception of customers in relation to a product depends on the elements taken into account during its manufacture. Bowman [34] asserts that the value capture is the combination of available resources and the ingenuity that we add to it. Those hypotheses will be verified on page 17.

### 4. Data and Methodology
*4.1. Data*

The data in this research is related to Samsung and Apple smartphones, covering the period 2011 to 2021. The data were collected from websites.

We have twelve variables in total. Among these variables, three relate to big data. Those variable are: active Facebook users, active YouTube users, and active Twitter users, over the period 2011 to 2021. Six variables relate to value creation: battery capacity, camera, CPU speed, RAM, screen size, and storage of the phone. The last three variables are related to value capture: the profits of the phone companies, the satisfaction of customers, and the unit of phones sold.

For big data, we selected the top three social media platforms used by the companies for their advertisements. We considered actives users of those social media to describe big data. We obtained these data from Statista website, and Meta's investor learning announcement web site. We did not use variables such as volume, velocity and variety as they are difficult to measure. Therefore, instead of those variables, we used the top three social media that companies used for their advertisement. Namely, Facebook, YouTube, and Twitter.

This is because those social media are associated with big data [34].

Regarding value creation, we focused on the hardware components of smartphones. Namely, we considered battery capacity, cameras pixel, CPU speed, RAM Memory, and screen size to describe value creation. This information was collected from the company's websites [43–45]. We could have considered more variables, such as variables related to software to describe value creation because various services can be provided to customer through software. However, in this article, we just considered the variables that we obtained during our investigation. For the same reason, three variables are considered to describe value capture regarding smartphones. We could have used more manifest variables in order to measure the value capture. The variables regarding value capture include profit, satisfaction, and unit of phone sold. is the data were collected from [43,46–49]. Profit is the gain made on a given smartphone. The profit is the difference between the cost of sale and the cost of production of the device. The satisfaction presents customers' reviews on amazon regarding devices. Indeed, on amazon we have the overall satisfaction for a given smartphone that corresponds to overall star rating of each smartphone. The star rating range between 1 star and 5 stars; 1 star means the customer is not satisfied and 5 stars means the customer is completely satisfied. The final score is the overall rating of the scores of all customers on amazon website. The unit sold is the number of smartphones sold during the first three months of its launch.

Partial Least Squares and Structural Equation Modeling (PLS-SEM) is the most appropriate methodology for this research. PLS-SEM is an approach that allows the estimation of a complex hierarchical model [7]. Indeed, the goal of this research is to bring out the relationship between big data, value creation, and value capture, on one hand, and the manifest variables that compose them, on the other hand. The most appropriate method when it comes to analyzing latent variables is PLS-SEM method [50,51].

*4.2. The Structural Model*

The structural model gives the structure of the model base on the latent variables. In the present research, "Big data" is an exogenous latent variable. "Value creation" and "Value capture" are the endogenous latent variables. Those variables are summarized by Equation (1). The relationship between those variables is described in Table 1.

$$Y = YB + Z \tag{1}$$

**Table 1.** The coefficients of Adjacency matrix D.

| LV | Big Data | Value Creation | Value Capture |
|---|---|---|---|
| Big data | 1 | 1 | 1 |
| Value creation | 0 | 0 | 1 |
| Value capture | 0 | 0 | 0 |

Equation (1) is the adjacency matrix. $Y$ is the latent variable matrix. In this paper, we considered "big data", "value creation", "and value capture" as latent variables (LVs). Among those latent variables, "Big data" is the exogenous variable, as shown in Figure 1.

Each coefficient indicates the connection between the variables. The arrows indicate the direction of the impact. The summary of the relationships between those latent variables is presented in Figure 1 and Table 1.

Table 1 is the adjacency matrix. It indicates the value of matrix $D$. If the value $d_{ij}$ = 1, the LV $i$ is then the predecessor of LV $j$. The matrix $D$ is similar to a triangular matrix, where $Y$ is the latent variables matrix. The error term $Z$ is assumed to be centered, i.e., [$Z$] = 0. The coefficients of matrix $B$ must be equal to zero where the matrix $D$ coefficients are zero. Therefore, we can formally write the equations (see Equation (1) for the model as follows):

*Big data = Big data + 0*

*Big data = Big data + 0 Value creation = $\beta_{12}$ Big data + $z_2$*

$$Value\ capture = \beta_{13}\ Big\ data + \beta_{23}\ Value\ creation + z_3$$

**Figure 1.** Structural model.

### 4.3. The Path Coefficients Calculation

Partial least square is used to calculate the coefficient of each path on the diagram of structural model. The path coefficients are calculated by applying the ordinary least squares method to each factor score of the structural model $\hat{y}_g$, $g = 1, \dots, G$. Where $\hat{y}_g$ refers to a given latent variable and G = 3 (3 latent variables). The path coefficients $\hat{\beta}_g$ is the coefficient of regression on its predecessor set $\hat{y}_g^{pred}$. The number of predecessors of each latent variable is indicated in Table 2.

$$\hat{\beta}_g = \left(y_g^{predT} y_g^{pred}\right)^{-1} y_g^{predT} \hat{y}_g$$

$$= Cor\left(y_g^{predT}, y_g^{pred}\right)^{-1} Cor\left(y_g^{predT}, \hat{y}_g\right)$$

where $T$ is the transpose matrix symbol.

**Table 2.** Predecessor of latent variable.

| Variables | Predecessors |
|---|---|
| Big data | 0 |
| Value creation | 1 |
| Value capture | 2 |

Table 2 indicates that "Big data" has no predecessor. "Value creation" has one predecessor, which is "Big Data". "Value capture" has two predecessors, which are "Big Data" and" Value Creation".

The elements $\hat{\beta}_{ij}$ $i,j = 1, \dots, G$, of B, which is the estimated matrix of the path coefficients.

$$\hat{\beta}_{ij} = \begin{cases} \hat{\beta}_j, for\ j \in \hat{y}_i^{pred} \\ 0\ else \end{cases}$$

### 4.4. The Whole Structure of the Model

The whole structure of the model is displayed in Figure 2. The model is composed of latent variables and manifest variables.

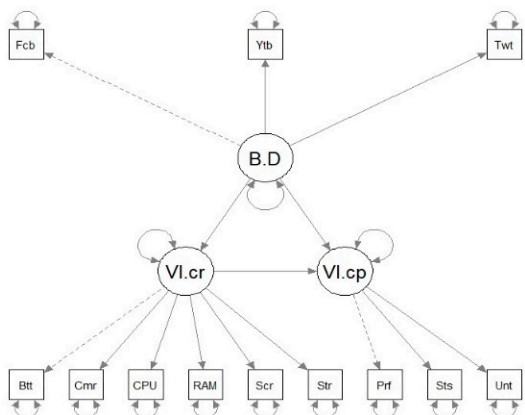

**Figure 2.** Structural equation modeling diagram.

For the purpose of representation, we have abbreviated the variables names. The latent variables are as follow: B.D. = "Big Data", V.Cr = "Value Creation", V.cp= "Value Capture". For the manifest variables relating to "Big data", Fcb = "Facebook", Ytb = "YouTube", Twt = "Twitter".

For the manifest variables related to "value capture", Btt = "Battery", Cmr = "Cameras", CPU = "Central Possessor Unit", RAM = "Random Access Memory", Scr = "Screen size", Str = "Storage".

For the manifest variable regarding "Value capture", Prf = "Profit", Sts = "Satisfaction", and Uni = "Unit sold".

*4.5. Measurement Model*

The measurement model is the model that relates (MVs) to their (LVs). Observed variables are referred to as MVs. LVs are seen as factors. As we mentioned previously, in the PLS-SEM framework, a given manifest variable can only be associated with one LV. Therefore, for our model, we obtain three latent variables which are: "Big Data", "Value Creation" and "Value Capture".

Big data. Three variables (volume, velocity, variety) are generally used to describe big data [36]. Volume indicates the amount of data collected or generated. Velocity is the speed at which data are generated. The velocity of data increases as time passes. Variety is the number of data types. The rise of new technology allows people to generate several types of data, such as text, photo, audio, video, and click-stream data. All types of data may be either structured, semi-structured, and unstructured. Recently, IBM added the fourth dimension, which is veracity. Veracity indicates in the data source whether there is unreliability and uncertainty in the data. SAS has also added two more dimensions to big data: variability and complexity. Thus, big data is characterized by manifest variables such as volume, velocity, variety, and variability. However, in this study, we will not use these variables to measure big data because it is difficult to measure those manifest variables. To measure big data, we will go through social networks users, namely Facebook, YouTube, and Twitter. The number of active users on those social networks gives an idea of volume, velocity of the data. This is because Samsung and Apple's favored social media are Facebook, YouTube, and Twitter.

Figure 3 displays the top three social media used by Apple and Samsung company for advertising. These social networks allow companies to present their products to the users of social network. The advertisements are in the form of image video, audio, text, etc.

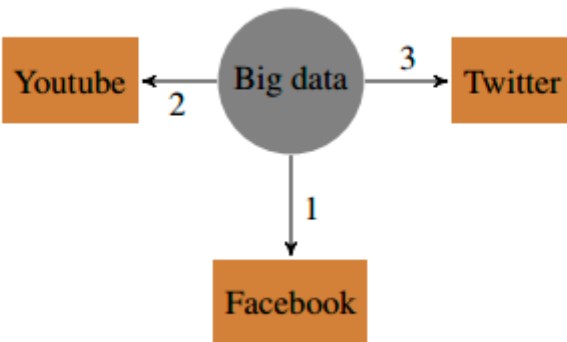

**Figure 3.** Top 3 social media advertising for Apple and Samsung.

Value creation. In the digital economy, one of the fundamental elements that creates value is undoubtedly digital data [37]. Value is created if, and only if, the sales value exceeds the cost of everything that is involved in creating that value. Value creation includes the following components: consumer surplus, profit margin, etc.

It is an execution of acts that increase the value of products, services, or even an organization, in a context of new technologies. Where information is increasingly available, business organizations are invited to expand their expertise area by creating new products related to new technology and services. In this research, we focus on the hardware components of smartphones to measure the value creation. To achieve our goal, we considered six manifest variables to describe "Value creation". Those manifest variables are: "Battery", "Camera", "CPU speed", "RAM", "Screen size," and "Storage". Those variables are represented in Figure 4.

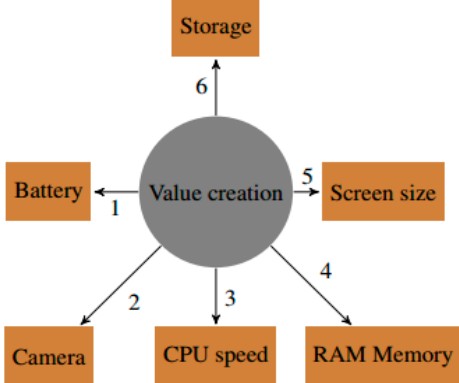

**Figure 4.** Value creation and capture.

Value capture. This is the capacity for organizations to preserve their surplus within their organizational limits. Capture value is becoming increasingly relevant, particularly in the digital economy. The value capture, in the age of the digital economy, is closely linked to digital data, innovation, as well as the interaction between customers. The exchange value is captured when it is sold.

E-commerce platforms themselves capture considerable value from trading, in the form of commissions or fees. Additional studies are required to understand how this commission is used and to understand how it varies in space and time, i.e., from one company to another and over the years. As part of this study, three manifest variables are considered to describe the value capture. These are profit, satisfaction, and the number of phone units sold. This is summarized in Figure 5.

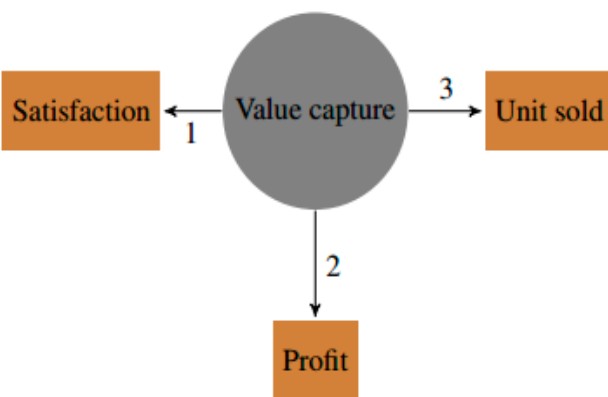

**Figure 5.** Value capture.

### 4.6. Partial Least Squares (PLS) Algorithm

According to Table 3, if the value $m_{kg} = 1$, the $MV_k$ is among the block of the latent variable $LV_g$. The purpose of the PLS algorithm is to estimate, by an iterative procedure, the values for the LVs. The objective is to build each LV by summing the MVs. Then in the inner calculation, we estimate each LV through its neighboring LVs. Table 4 contains the source and target variables for the measurement model. It indicates whether the latent and manifest variables are in formative or reflective way.

**Table 3.** Adjacency matrix M for the measurement model.

|  | **Big Data** | **Value Creation** | **Value Capture** |
|---|---|---|---|
| Facebook | 1 | 0 | 0 |
| YouTube | 1 | 0 | 0 |
| Twitter | 1 | 0 | 0 |
| Battery | 0 | 1 | 0 |
| Camera | 0 | 1 | 0 |
| CPU speed | 0 | 1 | 0 |
| RAM memory | 0 | 1 | 0 |
| Screen size | 0 | 1 | 0 |
| Storage | 0 | 1 | 0 |
| Profit | 0 | 0 | 1 |
| Satisfaction | 0 | 0 | 1 |
| Unit sold | 0 | 0 | 1 |

**Table 4.** Measurement model.

| **Source** | **Target** |
|---|---|
| 1. Facebook | Big data |
| 2. YouTube | Big data |
| 3. Twitter | Big data |
| 4. Battery | Value creation |
| 5. Camera | Value creation |
| 6. CPU speed | Value creation |
| 7. RAM memory | Value creation |
| 8. Screen size | Value creation |
| 9. Storage | Value creation |
| 10. Profit | Value capture |
| 11. Satisfaction | Value capture |
| 12. Unit sold | Value capture |

## 5. Results and Discussion

### 5.1. Application on Apple and Samsung Smart Phones

We first summarize the manifest variables that have been used. The data include 38 different types of iPhone and Samsung phones over the period 2011 to 2021. Table 5 is the descriptive statistics of the manifest variables that are used. Big data includes active Facebook users, active YouTube users and active Twitter users. The unit is in millions of active users. For value creation, the battery capacity is in mAh, the camera is in Megapixel (Mpx), the CPU speed in Gigahertz (Ghz), the Ram memory in Gigabyte (GB), the screen size is in inches and the storage of the phone is in Gigabyte (GB). For value capture, profit is in dollar, satisfaction is star rating, and the unit of phone sold is in million.

**Table 5.** Descriptive Statistic.

|  | Min | Max | Mean | Sd |
| --- | --- | --- | --- | --- |
| Battery | 1010.0 | 5000.0 | 2636.7 | 872.6 |
| Camera | 5.0 | 108.0 | 15.0 | 17.9 |
| CPU speed | 1.0 | 6.0 | 2.2 | 0.9 |
| RAM memory | 1.0 | 12.0 | 3.4 | 2.5 |
| Screen size | 3.5 | 6.9 | 5.3 | 0.9 |
| Storage | 16.0 | 512.0 | 88.8 | 105.2 |
| Profit | 313.0 | 958.5 | 514.4 | 137.1 |
| Satisfaction | 3.4 | 4.6 | 4.0 | 0.4 |
| Unit sold | 4.0 | 40.0 | 14.9 | 7.5 |
| Facebook | 1007.0 | 2910.0 | 1794.6 | 593.0 |
| YouTube | 700.0 | 2400.0 | 1428.9 | 505.6 |
| Twitter | 150.0 | 396.0 | 287.7 | 56.6 |

Figure 6 is the factor score of latent variables. Those latent variables are big data, value creation, and value capture.

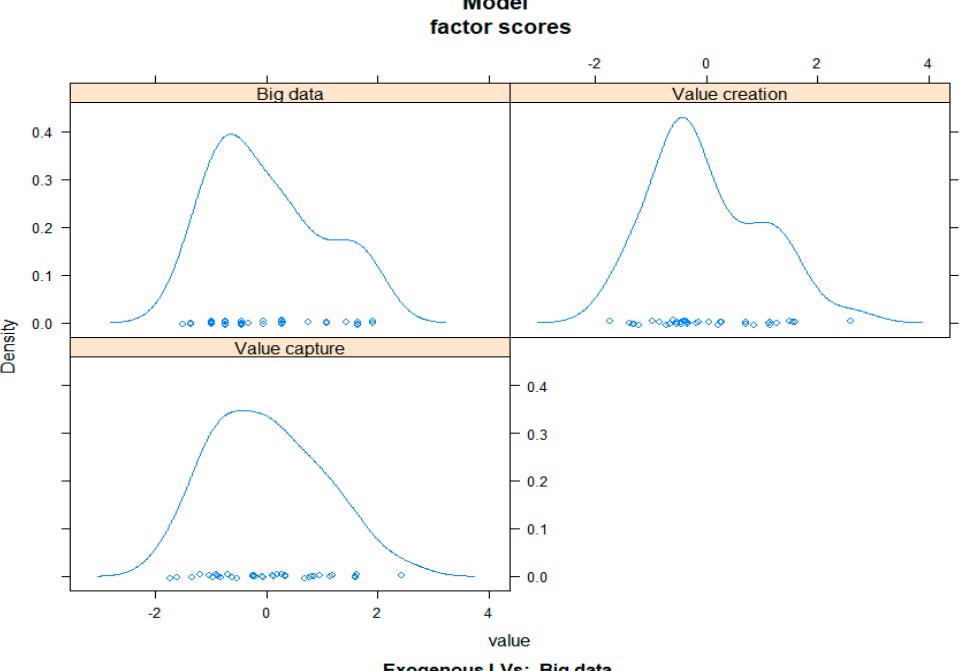

**Figure 6.** Factor Scores.

The mean of the factors score is equal to zero. The factor score is calculated based on the central limit law. The factor score or z score allows us to normalize all the variables and then facilitate analysis.

Figure 7 is the model prediction figure. It allows the prediction of value creation and value capture. Figure 8 is the model residual. Those residuals are supposed to be a normal distribution. This condition is necessary for the model. Table 6 shows the value of the total effect of latent variables on each other. In other word, the effect of big data on value creation and value capture, and of value creation on value capture.

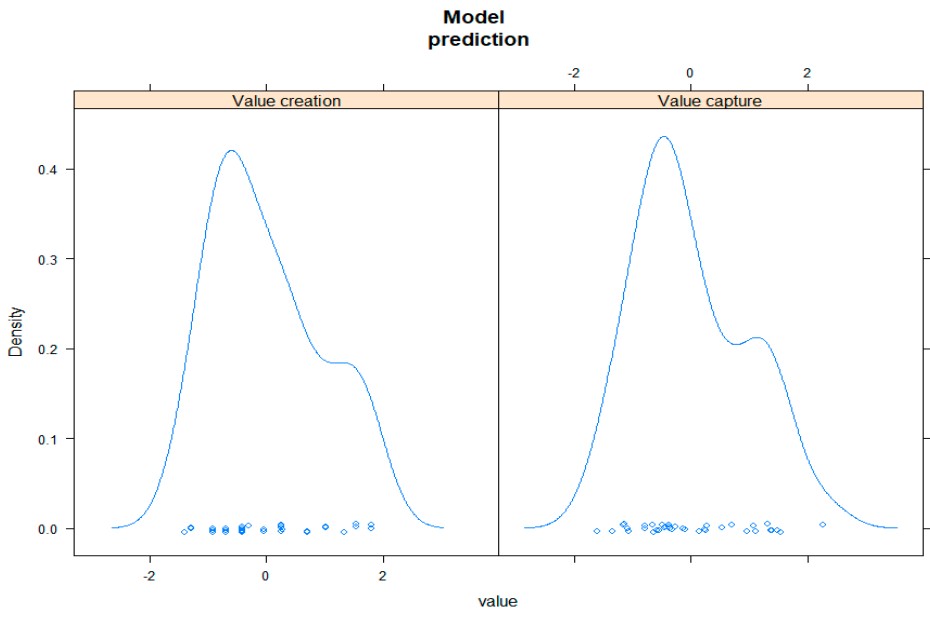

**Figure 7.** Model prediction.

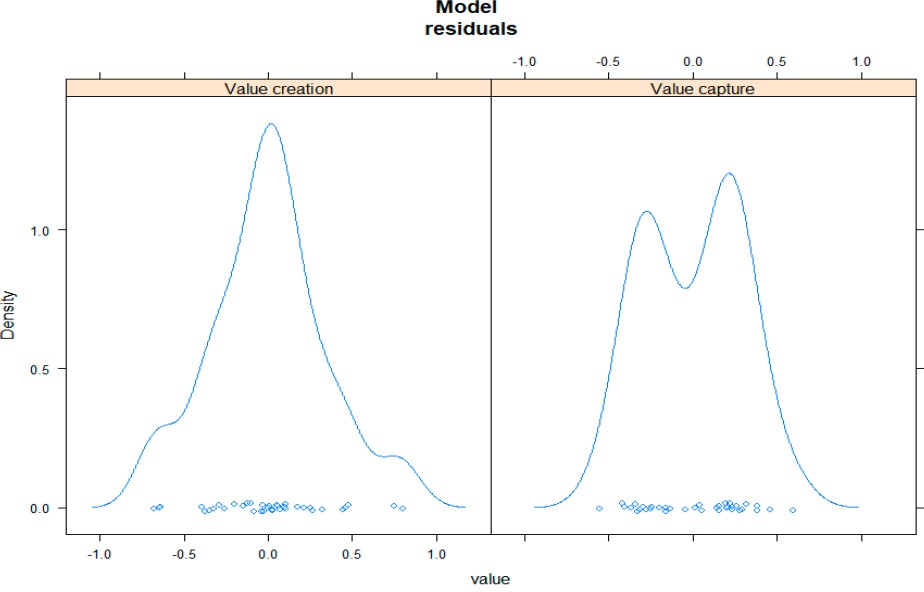

**Figure 8.** Model residual.

**Table 6.** Adjacency matrix D coefficients obtained after analysis.

| Variable | Big Data | Value Creation | Value Capture |
|---|---|---|---|
| Big data | 0 | 0.938 | 0.394 |
| Value creation | 0 | 0.000 | 0.578 |
| Value capture | 0 | 0.000 | 0.000 |

Figure 9 is the structural model with the estimated parameter values. The link between big data and value creation is 0.938; between big data and value capture, 0.394. The relation between value creation and value capture is 0.578. As those values are positive, that means there is a positive effect between the latent variables. Big data has great weight in terms of value creation and value capture. Similarly, value creation has a great effect on the process of value capture.

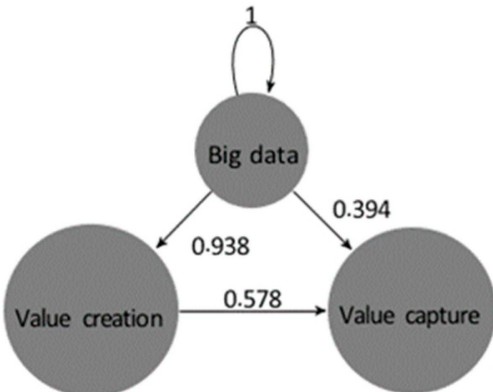

**Figure 9.** Structural model after analysis.

Table 7 indicates the value of the matrix for the measurement model. The table provides the weight of different manifest variables with respect to each latent variable. For "big data", we used active Facebook users, active YouTube users, and active Twitter users. The weight of each manifest variable in big data is, respectively, 0.994, 0.736, and 0.994. The coefficient associated with Facebook and YouTube is greater than that of Twitter. This means that these two variables have a large share in the big data of Apple and Samsung.

**Table 7.** Outer Weight.

|  | **Big Data** | **Value Creation** | **Value Capture** |
|---|---|---|---|
| Facebook | 0.994 | 0.000 | 0.000 |
| Twitter | 0.736 | 0.000 | 0.000 |
| YouTube | 0.994 | 0.000 | 0.000 |
| Battery | 0.000 | 0.719 | 0.000 |
| Camera | 0.000 | 0.385 | 0.000 |
| CPU speed | 0.000 | 0.871 | 0.000 |
| RAM memory | 0.000 | 0.827 | 0.000 |
| Screen size | 0.000 | 0.872 | 0.000 |
| Storage | 0.000 | 0.740 | 0.000 |
| Profit | 0.000 | 0.000 | 0.525 |
| Satisfaction | 0.000 | 0.000 | 0.860 |
| Unit sold | 0.000 | 0.000 | 0.920 |

For "value creation", the manifest variables are battery, camera, CPU speed, RAM, screen size, and storage. The correlation for all of the variables is positive. A positive value means that those variables positively impact the value creation. The variables that impact the value creation the most are CPU speed, screen size, and RAM. The relation between "value creation" is, respectively, 0.872, 0.871 and 0.827. Battery and storage also have great weight in terms of value creation. The associated coefficients are, respectively, 0.719 and 0.740. The camera does not have a great weight compared to other components. The associated coefficient is 0.385.

For value capture, profit, satisfaction of customers, and number of phones sold (unit sold) are the two variables considered. Unit of phones sold is the most important variable in terms of value capture. The weight of this variable is 0.920. The weight of satisfaction is 0.860 and the weight of profit is 0.520.

Table 8 is the R-square value for the partial least square model. R-square shows the quality indices and goodness of fit measures for the partial least square path models. Those values indicate that the model is well fitted. Big data has no predecessor; thus, it has no value estimated. The average R-squared is 0.89. Those values are calculated based on the predecessor of the latent variable, as mentioned in Section 4.3.

**Table 8.** R square of the latent variable.

|  | $R^2$ | $R^2$-Corrected |
|---|---|---|
| Big data | – | – |
| Value creation | 0.88 | 0.88 |
| Value capture | 0.91 | 0.91 |

The Figure 10 is the manifest variables pair of big data. On the diagonal, we have each manifest variable. The upper part of the diagonal is the correlation, and the lower part is the line of regression of the manifest variables, taken two by two. The correlation is 0.98 for YouTube and Facebook. For Facebook and Twitter, the correlation is 0.93. The value between YouTube and Twitter is 0.91. All the values of correlation are significant at 99%, as indicated by the three stars on the chart. Those values indicate the relationship between actives users on the different platform of social media.

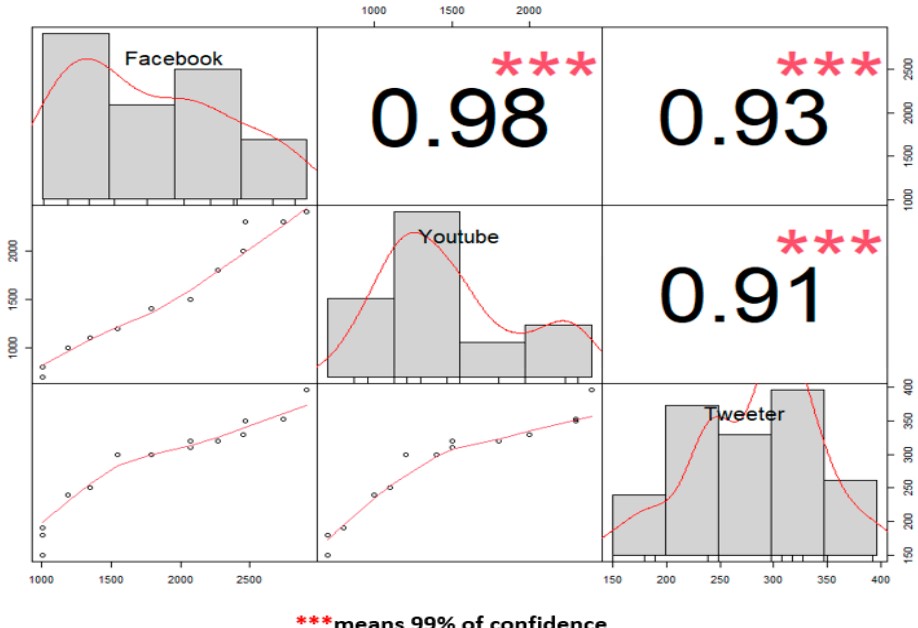

**Figure 10.** Big data.

Figure 11 is the manifest variables pair of value creation. On the diagonal, we have each manifest variable. The upper part of the diagonal is the correlation, and the lower part is the line of regression of the manifest variables, taken two by two. For example, the correlation is 0.90 between screen size and battery. This value indicates that there is a deep link between battery and screen size. For memory RAM and battery, the correlation is 0.83. The correlation between CPU speed and battery is 0.58.

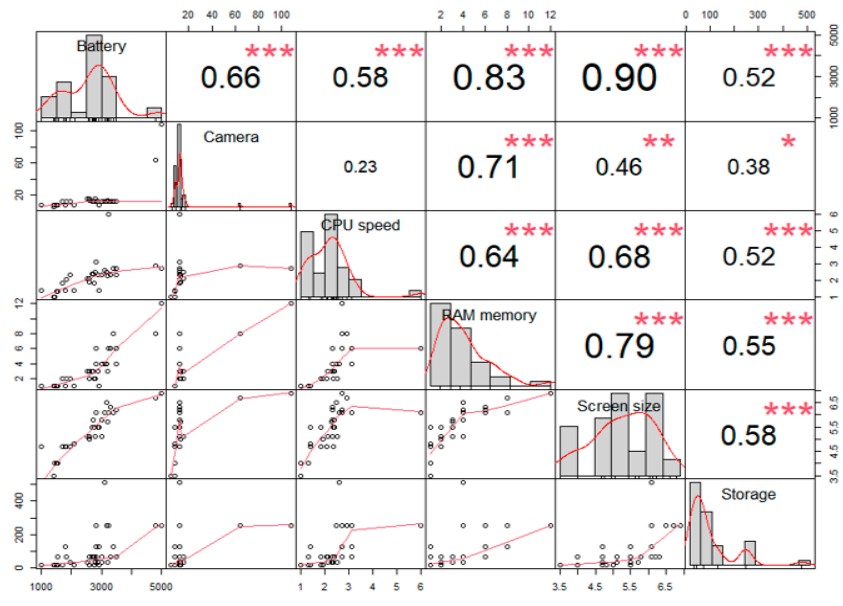

* means 90% of confidence; ** means 95% of confidence; *** 99% means of confidence

**Figure 11.** Value creation.

This means that there is also a relation between battery, CPU speed, and memory RAM. The correlation between camera and CPU speed is 0.23. Those values indicate the link between variables.

The Figure 12 is the manifest variables pair of value capture. On the diagonal, we have each manifest variable and the associated histogram. The upper part of the diagonal is the value of correlation and the lower part is the line of regression of the manifest variables. The correlation is 0.70 between satisfaction and units sold. This correlation is significant at 99%, as it has three stars. This means that the relationship between satisfaction and units sold is high. The correlation between units sold and profit is 0.35. The significant level is 90%. The link between profit and satisfaction is not significant; the value is 0.24. The Figure 13 displays the evolution of the algorithm iterations for each manifest variable of each latent variable. The figure is closely related to Table 7.

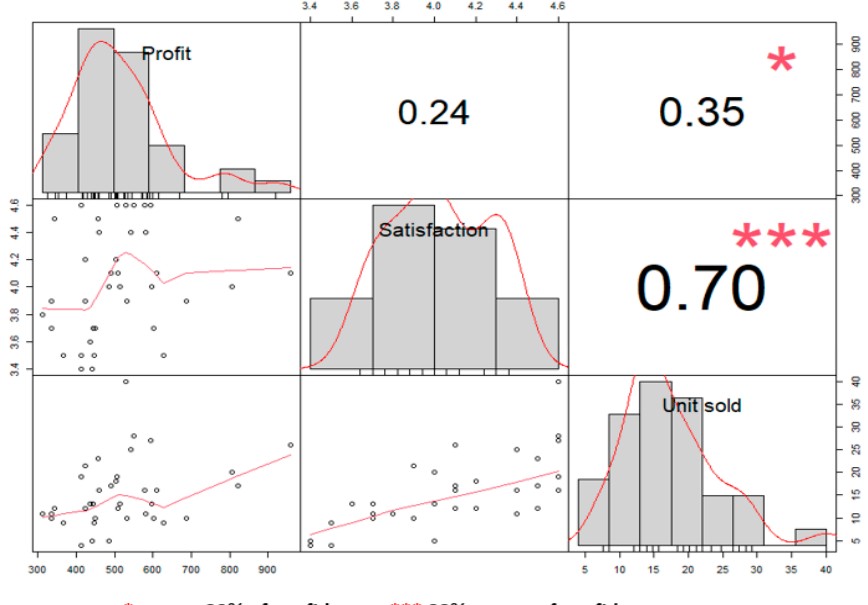

* means 90% of confidence; *** 99% means of confidence

**Figure 12.** Value Capture.

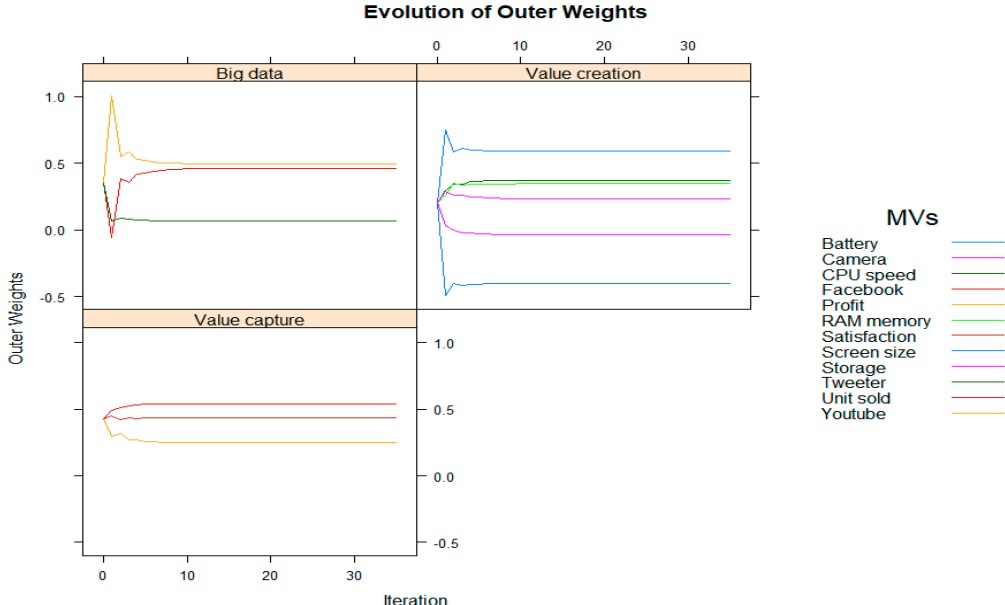

**Figure 13.** Outer weight.

### 5.2. Verification of the Hypotheses

We had made some assumptions regarding big data, value creation, and value capture on page 5.

To verify those hypotheses, we will refer to Table 9. Table 9 provides the estimate values of the path's coefficients, the bias, the standard error, and the confidence intervals of the estimate values. The significance level of the estimate values of the coefficients is fixed at 95%. The values in Table 9 are calculated using the bootstrapping method in the R software.

**Table 9.** R square of the latent variable.

|              | Estimate | Bias   | Std.Error | Lower  | Upper |
|--------------|----------|--------|-----------|--------|-------|
| $\beta_{12}$ | 0.938    | 0.017  | 0.013     | 0.927  | 0.979 |
| $\beta_{13}$ | 0.394    | 0.041  | 0.262     | −0.099 | 0.923 |
| $\beta_{23}$ | 0.578    | −0.045 | 0.260     | 0.033  | 1.060 |

The estimated value of $\beta_{12}$ is 0.939 with 0.017 std.Error. The confidence interval is [0.927, 0.979]. This means that the impact of "Big Data" on "Value Creation" is positive and different from zero. This result is statistically significant at 95% of confidence. In other words, the real value of $\beta_{12}$ is different from zero, with 95% of confidence. The first hypothesis is verified. There is a positive relation between "Big Data" and "Value Creation". "Big data" involves more effective smartphones (**Hypothesis 1**).

The estimate value of $\beta_{13}$ is 0.394 with 0.041 std.Error. The confidence interval is [−0.099, 0.923]. This means that the impact of "Big Data" on "Vale Capture" is positive. This result is not statistically significant at 95% of confidence. Indeed, the confidence interval contains zero. This means the real value of $\beta_{13}$ might be zero. However, the result is significant at 80% as the confidence interval is [0.134, 0.753]. Therefore, we can conclude at 80% there is positive relation between "Big Data" and "Value Capture" (**Hypothesis 2**).

The estimate value of $\beta_{23}$ is 0.578 with 0.260 std.Error. The confidence interval is [0.033, 1.060]. This means the impact of "Value Creation" on "Value Capture" is positive. This result is statistically significant at 95% of confidence. This is because the confidence interval does not contain zero, meaning that the real value of $\beta_{23}$ is different from zero.

This means the more value we create, the more we capture. Value creation has great impact in term of value capture (**Hypothesis 3**).

*Discussion.* Sustainable development can be an engine that improves a company's competitive strategy and the development of new business opportunities. Sustainability highlighting the value of co-creation through multi-stakeholder partnerships. The focus of this section is to point out the role of big data in value creation and value capture in smartphone companies that leads to the sustainable development of their activity. Smartphone manufacturers are in competition when it comes to selling their products. To stay the course, they act on the characteristic elements of their product in order to meet the needs of their customers and thus bring satisfaction. This makes it possible to retain a larger target population [16]. The industrial value of this research is to bring out the elements on which it is necessary to act when one wishes to bring innovation. The main result will focus on this aspect.

The results show that big data has a significant impact on value creation. More specifically, we found that big data has a positive effect on value creation regarding smartphone features in Apple and Samsung. The result is significant at 95%. Our finding is consistent with previous scholars. Indeed, Ref. [8] conducted a study in nine companies using big data technology. They found that big data helped create a lot of value. They explain this by the fact that big data has brought in new skills, such as the data scientist, who can dig into the data and extract the information needed to create value. He did not give an exact figure, but claims in his results that big data has a positive impact on value creation. In our context, big data from social media subscribers' comments helps phone companies to add some features to phones that will increase the value of the phone and then attack more consumers. Singh et al. [7] studied the effect of big data on sustainable capacity building. They have found that big data allows companies to create value and thus allows firms to be more competitive. Similarly, Dekimpe et al. [34] assert in their results that big data makes it possible to create value and to appropriate value. In view of these results, we claim with confidence that big data is an important lever that can boost the creation of value in smartphone companies and, therefore, add value to their economy.

Regarding big data and value capture, we note that big data can help to capture value in a significant way. Big data allows companies to sell large numbers of phones. The more data we have, the more phones we sell. This is confirmed by our results. Indeed, our results indicate that capturing value is strongly associated with big data for Samsung and Apple. Big data helps phone companies to sell a lot of phones. This result is significant at 95% of confidence. This association between big data and value capture has already been proven by Olszak et al. [33] in their study on the creation of value in companies.

The result on the relationship between value creation and value capture is also conclusive and significant at 95% of confidence. Our findings indicate that for Samsung and Apple, value capture is strongly linked to value creation. Indeed, the more Samsung and Apple manufacture smartphones with good characteristics, the more they sell and the consumers are satisfied. This result is confirmed by Pitelis [52]. He finds that there is a positive effect of value creation on value capture. His result is in agreement with our result because value creation impacted positively on value capture, even if he does not give an exact figure, as we also found that value creation has a positive effect on value capture.

## 6. Conclusions

In this academic research, we addressed value creation and value capture with big data in smartphone companies in an era when social media are generating huge amounts of data. The study focused on the role of big data in the value creation and value capture from Samsung and Apple phones. To achieve our result, we considered three blocks of manifest variables. Each block represents latent variables, namely, "Big Data", "Value Creation", and 'Value Capture". The manifest variables related to big data are actives users on the top three social media used by the companies for advertising, namely, Facebook,

YouTube, and Twitter. For value creation, we took into account battery, CPU speed, RAM, storage, screen size, and camera. Profit for phone firms, customer satisfaction, and units of phones sold are the three manifest variable that we considered for value capture. Structural equation modeling and partial least squares are the statistical tool we used to obtain the result. The results show that the CPU speed, RAM, and screen size are principally what interest customers in term of value creation. Thus, we suggest smartphone companies focus more of their research and development on the value creation related to the speed of the phones and everything that is related to the speed of the phones and batteries capacity. This would allow the companies to capture more value in terms of profit, and also satisfy customer needs. Active Facebook users, active YouTube users, and active Twitter users are observed variables that impact "Big data", in this order. We obtained the estimated values that measure the relationship between different variable considered in this paper. "Big data" positively impacts "Value Creation" and "Value Capture". Similarly, "Value Creation" is positively correlated to "Value Capture". To verify this, we assumed three hypotheses that figure out the relation between "Big Data", "Value Creation", and "Value Capture". Among those hypotheses, two are significant at 95%. Therefore, we conclude that "Big Data" is essential in terms of value creation because it positively impacts "Value Creation". Similarly, "Value Creation" is close related to "Value Capture", at 95%. The last hypothesis is not significant at 95%. However, we can conclude that "Big data" positively impacts "Value Capture", at 80%. Future research with more data to confirm these results is encouraged.

In view of these results, the importance of big data in terms of creating and capturing value in smartphone companies remains necessary. Big data technologies are an important lever for companies, as indicated by our findings. Integrating big data technology into companies and firms remains an essential choice to optimize the profitability of these firms. Indeed, big data makes it possible to analyze the complex data of users of social networks. At present, with big data technology, companies can analyze the data and understand the behaviors and feelings of users towards their products. The results of this analysis can help companies in their decision-making regarding the manufacture of smartphones. Good decision-making allows companies to adjust the manufacturing and cost of phones to optimize the final cost of the product. This will allow a balance between companies and consumers in terms of benefits. We recommend, similarly to the previous researchers regarding the effect and share of big data in the economy development, the integration of this technology in companies in order to take full advantage of it in the era of ICT (Information and Communication Technology), where data explodes. The data from social networks not only make it possible to create value, but to capture it. As a result, we see that big data has a double effect on value capture. It has a direct effect and an indirect effect. The direct effect of big data is on value creation and the indirect effect of big data is on value capture. Big data positively impacts the creation of value. Indeed, with big data technology, Samsung and Apple manage to adjust the characteristics of their mobile phone to satisfy their customers. The creation of value is conducted through the improvement of the characteristics of smartphones, such as their batteries, the size of the screen, the processor, and the RAM (Random Access Memory) because most customers pay special attention to smartphone features [53]. Captured value is realized when companies manage to sell their products to make a profit and when customers are satisfied with the product purchased. This is where the industrial value of this research takes on its full meaning. Indeed, big data gives a direction to follow in terms of innovation.

This study has a limit. We faced a data collection problem. Indeed, we have scrapped data from websites. The data we obtained were not large. For future studies, it is better to address such research by collecting more data from all telephone companies in order to obtain an accurate result. We failed to take into account all the manifest variables that describe "Big Data", "Value Creation", and "Value Capture". We invite future researchers to add some manifest variables, such as variables related to the software of smartphone, in order to describe the value creation while addressing this topic.

**Author Contributions:** P.L.: Methodology: Supervision. M.A. (Maniyassouwe Amana): Conceptualization, Data curation. M.A. (Mona Alariqi): Resources. All authors have read and agreed to the published version of the manuscript.

**Funding:** This research received no external funding.

**Institutional Review Board Statement:** Not applicable.

**Informed Consent Statement:** Not applicable.

**Data Availability Statement:** Not applicable.

**Conflicts of Interest:** The authors declare that the research was conducted without any commercial or financial relationships that could be construed as a potential conflict of interest.

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
