# Peer review of "Value Creation and Capture with Big Data in Smart Phones Companies"

_sustainability, doi:10.3390/su142315882_

Round 1

Reviewer 1 Report

The paper, Value creation and capture with big data in smart phones companies, addresses a research area interesting in marketing (?) (consumer’s satisfaction) and big data, however, sustainability context was not inserted... In addition, some aspects should be considered. For example, the Introduction section should be more shortened, where the authors could address that has been studied previously or is not known? The research gap and the main objective this study should presented clearly. The introduction should clearly indicate the need for this paper in relation to extant research studies and its importance in sustainability area.

The methodology needs further development to indicate its validity and reliability. The findings are very general and thus the implications are very general as well. The authors draw on several prior studies, but a much more critical literature analysis is needed to strengthen the paper’s argument and draw out the gaps they seek to address. Which gap(s) in extant studies are authors trying to address here?   While the author(s) establish some links to some literature, the authors need to establish a more explanation for framework and research hypotheses.

The paper presents a structure complex. For example, the section 3 (Data) should be inserted in Methodology section.

Also, the paper needs to be present much stronger results discussion and conclusion sections in order to offer value to the reader. While you have made a valiant attempt to tackle these topic, I would argue that the data used is are “good”. However, a more discussion is need focused in sustainability issues… There are a number of very findings that make it unwieldy for a smooth reading…. The paper lacks an appropriate methodological procedures hence the text tends to wander and even approach waffle at certain points…

The conclusions and implications could be extended, innovative and more contributions for theory and practice should be presented.! Some limitations and further research agenda should be also presented.

Author Response

Hello professor. Thank you very much for your comments. In the revised article, 

1) Introduction has been shortened to present clearly the research gap and the main objective.  The introduction also indicates the need for this paper

2) Methodology section is detailed now clearly to indicate its validity.  

3) The results discussion is revised to offer value to the reader as you suggested.

4) The conclusion is extended and more explain.

Reviewer 2 Report

My comments and questions are the enclosed paper.  I did have a few questions, but my main concern is one statement, "They found that big data helped create a lot of value." I don't believe that big data creates value or impacts value creation.  Rather, I  believe that big data can give us a  better and more accurate estimate of what that value is. 

Author Response

Thank you very much for your comments. Regarding your question of whether big data creates value. The answer is yes. the objective of the research is to show how big data can help create value. Indeed, big data contains enough information from commentators on the networks. Consumers give their points of view in relation to smartphones. The more data we have, the more information we have about the consumers of the products. These insights and comments on social media help make good decisions about innovation. This creates value.

Reviewer 3 Report

The theme is worthy of investigation. However, the following needs to be addressed. I read the manuscript, I found some areas in which I would have appreciated greater clarity. I believe the paper could be further strengthened by added information.

1. The research significance, including theoretical and practical significance.

2. The urgency of the study should be stated in the abstract.

3. The findings are not reflected in the abstract.

4. Literature from the last three years should be used.

5. The introduction suggests stating the research method and its advantages and disadvantages.

6. The introduction suggests studying the impact of big data on value creation and capture, indicating research significance.

7. The introduction points out that the research mainly focuses on the framework and theory of big data, its application and its 40 dimensions, and does not specifically reflect the research gap. It is suggested to add the literature to prove it.

8. Variable selection should be added to the literature evidence.

9. Full text modification format.

Author Response

Thank you very much for your comment.

1) The research significance  is added to the main objective of the paper in the introduction

2)  This research is important and urgent. Indeed, in the era of social networks, data has become raw material for decision-making. Therefore, it is important and urgent to study the relationship that exists between data and the creation of value. It is in this perspective that we make this study in order to understand the relationship

3) The abstract is reformulated differently so that it reflects the results

4) Literature from the last three years is used. Most of the articles cited here are recent. With the document format, you can see it directly in the literature. 

5)  We have added the advantages and disadvantages of the methods used in the introduction.

6) The research significance is added in the introduction.

7)  The research gap is specified in the introduction.

8 Variable selection is added in the literature 

9 ) I have changed the text format

Round 2

Reviewer 1 Report

The authors insert not my suggestions along the paper.

Author Response

Dear reviewer, Thank you very much for your comments. I reviewed my paper now according to your comments. I highlighted the part I changed in the paper. Please check it in the Abstract, Introduction, Literature review, discussion, and conclusion

Reviewer 3 Report

The authors completely failed to revise the manuscript in accordance with the previous comments. I don't see any changes. In particular, the literature review of this manuscript is extremely lacking. I'd like to give the author another chance to revise. Otherwise, I have to reject the manuscript.

Author Response

(The authors gave the same response as above.)

Round 3

Reviewer 1 Report

The authors inserted some of my suggestions.

Author Response

Thank you dear reviewer for your comments. I revised my paper once again to meet your requirements. The parts I  added are highlighted in green Color. In the introduction. From line 84 to line 89. Line 92 to line 94. In the discussion from 681 to 685. In the conclusion from lines 760 to 761.

Reviewer 3 Report

It has enlarged explanations, but the main points remain: the numbers have no real-industry documentation, thus the example has didactical value mainly.

Author Response

Thank you dear reviewer for your comments. I revised my paper once again to meet your requirements. The parts I added are highlighted in green Color. In the introduction. From line 84 to line 89. Line 92 to line 94. In the discussion from 681 to 685. In the conclusion from lines 760 to 761